# The Roles of TP53 and FGFR2 in Progress Made Treating Endometrial Cancer

**DOI:** 10.3390/diagnostics12071737

**Published:** 2022-07-18

**Authors:** Olga Adamczyk-Gruszka, Agata Horecka-Lewitowicz, Agnieszka Strzelecka, Monika Wawszczak-Kasza, Jakub Gruszka, Piotr Lewitowicz

**Affiliations:** 1Department of Gynecology and Obstetrics, Collegium Medicum, Jan Kochanowski University, 25-369 Kielce, Poland; 2Department of Obstetrics and Gynecology, Province Hospital, 25-369 Kielce, Poland; 3Institute of Health Sciences, Jan Kochanowski University, 25-369 Kielce, Poland; agatalewitowicz@gmail.com (A.H.-L.); astrzelecka@ujk.edu.pl (A.S.); 4Department of Surgical Medicine with the Laboratory of Medical Genetics, Institute of Medical Sciences, Jan Kochanowski University, 25-369 Kielce, Poland; mwawszczak@ujk.edu.pl; 52nd Department of Obstetrics and Gynecology, Medical University of Warsaw, 02-091 Warszawa, Poland; gruszkajakub65@gmail.com; 6Department of Clinical and Experimental Pathology, Institute of Medical Sciences, Jan Kochanowski University, 25-369 Kielce, Poland; lewitowicz@gmail.com

**Keywords:** endometrial cancer, microsatellite instability, immunohistochemistry, illumina sequencing

## Abstract

The morbidity and mortality caused by endometrial cancer (EC) is still rising worldwide. In recent years, a new system of tumor stratification has been proposed based on POLE-mutational status, TP53, and microsatellite stability status. The aim of the study was to analyze a vast panel on the genes potentially involved in the genesis of endometrial cancer in the Polish population. One hundred and three white female patients with confirmed endometrial cancer were enrolled on the study. We performed sequencing using the Hot Spot Illumina panel and microsatellite stability with immunohistochemistry. We confirmed a key role of the TP53 mutation in progress to high-grade EC and parallelly some role of FGFR2 mutation. Moreover, our data present a vast landscape of mutations in EC and their polymorphism. We reported the meaning of FGFR2 mutation and TP53 (high copy number) in high-grade ECs. Our observation in MSI contribution is comparable with other studies. Finally, we see a strong need for the implementation of the TCGA classification.

## 1. Introduction

Endometrial cancer (EC) is considered the sixth most common malignant neoplasm in women. According to global statistical data, it is the second most common cancer of the female genital organs after cervical cancer. The highest incidence of EC is observed in well-developed countries in North America and northern and eastern Europe [1,2]. In the United States and Canada, EC is the most common gynecological cancer, while in China, EC is in the second position after cervical cancer [2,3,4].

The morbidity ratio of EC is increasing in 26 countries worldwide, with the largest increase in Asia and Africa, largely due to their rapid socio-economic changes. In the United States, EC is one of the few malignant neoplasms with increasing incidence and mortality. Between 1999 and 2016, the incidence of EC increased by 0.7% per year, while the mortality rate increased by 1.1% per year [5]. Breast cancer is the most common female malignancy in Poland, and EC frequency still raised over the past four decades. According to the statistical data for 2010–2017, the highest number of cases of endometrial cancer in Poland involved women in the 60–64 age group, and the highest number of deaths caused by this cancer was recorded in the 75–79 age group [6]. That EC overall survival dropped from 86.9% to 82.7% (*p* < 0.05), and from 2019 to present to 81.2% [3,7]. Endometrial cancer mainly affects older women, but there are cases in women younger than 40 years old. In most cases, it is diagnosed in the early stage of clinical advancement, for whom the five-year survival ratio is about 95%, in contrast to endometrial cancer detected in late-stage, where survival ranges from 16% to 45% [8,9]. The classical division of endometrial cancer includes two distinct pathogenetic subtypes that differ in histological and molecular characteristics, as defined by Bokhman [8].

Type I (endometrioid adenocarcinoma) accounts for 80–90% of cases, and type II (non-endometrioid adenocarcinoma) includes serous carcinoma, clear cell carcinoma, undifferentiated, and mixed tumours.

Type I is based on the previous atypical endometrial hyperplasia showing estrogen-dependent proliferation and a high ratio of PTEN mutation [2,10,11]. These tumors are considered as less aggressive, often found in the early stage according to FIGO in stage I and II, with high positive steroid receptor status as an indicator of a favorable prognosis [12].

The less common type II, usually without atypical hyperplasia of the uterine mucosa, most often in postmenopausal women, occurs in 10% of patients with EC, and contains high-grade lesions of serous and clear cell carcinoma driven by TP53, KRAS mutation and a high ratio of HER2 amplification [1,2,3,4,5,6,7,8,9,10,11,12,13].

Many studies have identified PTEN mutation as the most frequent molecular event causing the initiation of intraepithelial neoplasia. Animal models of endometrial cancer have demonstrated that biallelic PTEN loss leads to the development of atypical hyperplasia, whereas biallelic PTEN loss, together with the PIK3CA mutation, leads to cancer. These findings confirmed the fact that PTEN mutation commonly co-occurs with PIK3CA and PIK3R1 mutation. A deep molecular insight into causative pathways allowed to separate the EC subgroups according to molecular pathway and clinical outcome. The first subgroup covers the POLE-mutated tumors. Patients in this subgroup presented the best disease-free survival (DFS). In the second subgroup are tumors which additionally presents microsatellite instability (MSI) or MLH1 promoter hypermethylation. The third subgroup includes low copy number (CNV)/and microsatellite stable (MSS) tumors with frequent CTNNB1 (Wnt-pathway) mutation. Finally, women in the fourth subgroup were characterized by high CNV via frequent TP53 mutation. In this subgroup, overall survival (OS) and DFS are poor [14].

The molecular subgroup proposed by the Cancer Genome Atlas (TCGA) has evolved the two-tier Bokhman’s model into a precise molecular-based system [15]. This has resulted in the creation of a new diagnostic algorithm and clinically meaningful EC classification. The pathology reports should include information about the mutational status of the POLE gene and perform immunohistochemical p53, MSH6 and PMS2 to provide an integrated ‘histo-molecular’ diagnosis [16]. Recent papers confirm the diagnostic value of immunohistochemistry in MSI screening. In addition, the MSI status is now recognized as the very promising therapeutic target in advanced EC given the initial results of immune checkpoint inhibitors. Clinical trials evaluating these drugs in the early stage will start soon [17,18].

We aimed to analyse a vast panel on genes potentially involved in the genesis of endometrial cancer in the Polish population.

## 2. Materials and Methods

### 2.1. Study Population

One hundred and three white female patients with confirmed endometrial cancer were enrolled in the study. All patients were operated on in Province Hospital, Kielce, between 2005 and 2017. A total hysterectomy with pelvic lymphadenectomy was performed. The collective data for evaluation of follow-up were tabulated and successively enhanced annually in all cases. (Table 1) A computer tomography examination of the abdomen and chest, a typical blood test, and an endoscopic examination were performed. Each case was re-diagnosed according to the Eighth Edition of TNM Classification [17]. 

Importantly, all of the studied participants underwent typical surgical treatment without previous radio-chemotherapy. This resulted in a credible comparative analysis of tumor characteristics in the scope of treatment and an unchanged molecular profile.

### 2.2. Microsatellite Stability

Microsatellite stability status was assessed by immunohistochemistry. The immunohistochemical assays were performed using the automated IHC/ISH slide staining system BenchMark Ultra (Ventana Medical Systems; Roche Group, Tucson, AZ, USA). A complete diffuse lack of positive nuclear reaction of MLH-1, MSH2, MSH6, and PMS2 was coded as positive MSI.

### 2.3. Molecular Analysis

DNA isolation: Cancer genomic DNA was extracted from formalin-fixed paraffin-embedded (FFPE) tissue using the MagCore^®^ Genomic DNA FFPE—Step Kit (RBC Bioscience, New Taipei City, Taiwan). The quality was quantified using DeNovix DS-11 Spectrophotometer (DeNovix, Wilmington, DE, USA) and QuantiFluo^®^ ONE dsDNA System (Promega, Madison, WI, USA).

Library preparation for NGS analysis: The analysis of genes involved in cancer was performed using the amplicon-based method AmpliSeq Cancer HotSpot Panel v2 for Illumina (San Diego, CA, USA). List of analyzed genes is presented in Appendix A

Amplicon-based gene panel protocol: Amplification of the hotspot regions of 50 oncogenes and tumour suppressor genes was carried out using AmpliSeq™Library PLUS for Illumina^®^ assay Kit (San Diego, CA, USA) according to the manufacturer’s instructions (AmpliSeq for Illumina Cancer HotSpot Panel v2 Reference Guide). Briefly, the assay generated a library of 207 gene-specific amplicons and targets ~2800 clinically relevant mutations. The amplification was performed in two rounds of PCR: Initially, 72 ng of DNA was used for each sample to perform multiplex PCR reactions that amplified the entire target regions. The adapters ligation was performed using AmpliSeq™ CD Indexes Set A for Illumina^®^. The small residual DNA fragments were removed by use of a magnetic-based DNA purification approach. The product of each sample was used as a template for the second amplification step, which amplifies the product with sequencing primers. Finally, each tagged amplicon library purification was performed using NucleoMag^®^ NGS Clean-up and Size select beads (Machery-Nagel GmbH& Co., Düren, Germany). Each library was qualified using QuantiFluo^®^ ONE dsDNA System (Promega, USA) to allow for the equimolar pooling of all sample libraries for subsequent sequencing.

Sequencing: Products were analysed by Next Generation Sequencing (NGS) using the Illumina platform MiSeq Dx. Briefly, NGS was performed using a MiSeq Reagent Micro Kit v2 (300-cycle) (Illumina, San Diego, CA, USA). Indexed DNA library concentrations were quantified as described above and normalised to 2 nM using Low TE. The library was denatured using 5 μL of 2 nM library and 5 μL 0.2 N NaOH. The library was diluted using pre-chilled HT1 buffer at a final concentration of 20 pM. Finally, the 8 pM library was spiked in 5% of PhiX Control v2 (Illumina, San Diego, CA, USA), which provides quality control for cluster generation, sequencing, and alignment.

Data analysis: Analysis of NGS data was performed using the GALAXY platform (usegalaxy.org). Sequencing reads (FASTQ files) were aligned to the human reference genome hg19 using the Bowtie2 tool. Variant calling was performed using the Varscan2 tool. The parameters used for data analysis were minimum allele frequency, 0.05; minimum quality, 20; and minimum coverage ×80. All called variants were annotated using wANNOVAR (https://wannovar.wglab.org (accessed on 10 February 2019)). Results were visualized using the R Bioconductor package Maftools (http://bioconductor.org/ (accessed on 10 February 2019)).

### 2.4. Statistical Methods

Quantitative data were reported as mean and standard deviation or median and range. Categorical data were expressed as number and percentage distributions. The Chi-square test or Fisher’s exact test was applied to compare proportions, and a multivariable logistic regression model was used for assessing the relationship between the targeted genes. Follow-up time was calculated as the number of years from surgery to disease recurrence, death from reasons other than cancer, or the last contact with the patient. The univariate associations between disease-free survival and selected patient and tumor characteristics were evaluated by the univariate Cox proportional hazard model, and for those analyses, continuous variables were dichotomized according to the median. To identify the independent prognostic factor for disease-free survival, a multivariate Cox proportional hazard model with backward selection (with cut-off 0.05) was performed on only those statistically significant variables in univariate analysis.

All statistical tests were two-sided, and values < 0.05 were considered significant. Computations were performed using STATISTICA (data analysis software system), StatSoft, Inc., Palo Alto, CA, USA (2014), version 12, http://www.statsoft.com/ (accessed on 10 February 2019).

## 3. Results

### 3.1. Clinical Characteristics of the Study Population

A total of 103 woman aged from 48 to 99 years old with different type of endometrial cancer and full clinical history were included in this study. It was observed that the largest group were woman between 61–80 years old (62%). It was also noticed that 97% of women in the study group were obese or overweight (BMI > 30). The average time of the overall survival (OS) was 8.23 years. The time of relapse-free survival (RFS) was recorded only in six cases, and the average time was 4.6 years. The detailed description of the study group is presented in Table 1 and Appendix A. Additionally the characterization of Histopathological types of EC is presented in Appendix A.

### 3.2. Microsatellite Stability Status

In the studied cohort, we noted a 28% frequency of microsatellite instability. The lack of MLH-1 was observed 13.5% cases, MSH-2 in 1 case, MSH-6 in two cases, and PMS-2 in 8.7% of cases. MSI and especially MLH-1 silencing positively correlated with RFS (*p* < 0.01). Moreover, all MLH-1 positive tumors lacked MLH-1 mutation in the used molecular panel, and moreover were connected with high grade EC (*p* < 0.01). Interestingly, in a multifactorial analysis, MLH-1 followed the FGFR-2 mutation and led to higher tumour grade (*p* < 0.01).

### 3.3. Genotyping

All results were analyzed in terms of quality. All nucleotides were covered with at least the minimum amount of reads necessary for reliable variant detection with the modified targeted capture-based approach in 67 cases. Thirty-seven cases were considered low-covered following the generally applied minimum read count for nucleotides in the targeted capture-based approach. These cases were excluded from the analysis. Generally, in the studied cohort, mutations in 14 analyzed genes were detected. Figure 1 depicts a characteristic of the mutational spectrum. Additionally the mutation spectrum according to histopathological type and grade is presented in Appendix A.

We noted the missense mutation as the most frequent single nucleotide polymorphism (SNP). PTEN mutation was found in 49% of cases, and interestingly, here, we observed a marked heterogeneity of mutation variants. Exon 5 codon 130 (c.C388G) was the most frequently altered. As was presented in Figure 2, we noted that many cases harbor more than one mutation.

### 3.4. Genetic Background and Clinical Implications

The statistical analysis did not reveal a clear dependence of mutation type with tumor grade, RFS, and OS (*p* > 0.05). One case of clear cell carcinoma presented no mutation in the user panel. Multivariate analysis showed the dependence of the tumor dedifferentiation with FGFR2 and TP53 (*p* < 0.01). Moreover, the FGFR2 mutation was more frequently observed in more advanced tumors (*p* < 0.01). We performed the analysis covering the number of SNP in aspect of RFS, OS, and age. The correlation was not statistically significant. However, the TNM strongly correlate with the TP53 SNP number (*p* < 0.01).

It is worthy of note that the FGFR2 and TP 53 mutations correlated well with tumor grade progress. Tumor stage progress depends on the FGFR2 mutation and the mutation number in TP53. Interestingly, one case of 59-year-old female (endometrioid carcinoma G2, pT1b) revealed six mutated genes (including KRAS, TP53, APC and CTNNB1) in 11 SNP, and after 14-years of observation, the patient is still alive. In our study, we found 10 cases of EC G1 with excellent follow up (up to 15 years) with no mutation in the user panel. The Kaplan-Meier analysis concerning TP53 (HR = 2.26; 95% Cl: 0.222-23.074; *p* = 0.067), MLH-1 (HR = 2.34; 95% Cl: 0.465–11.730; *p* = 0.303), MSI (HR = 0.55; 95% Cl: 0.155–1.933; *p* = 0.350), and FGFR-2 (HR = 0.40; 95% Cl: 0.142–1.144; *p* = 0.975) showed no significant results, but the impact of FGFR-2 to OS was more significant than TP53 (Figure 3).

Figure 4 presents a main molecular pathways involved in EC genesis and progression. Not surprisingly, PTEN mutation dominates, leading to AKT-mTOR activation, but in parallel, in 14% of cases, we observed an additional FGFR2 booster. A spectrum of TP53 mutations leads to an excess of misfolded p53 or a total lack of p53. Overstimulation via the Wnt-pathway and the β-catenin excess contributes to 12% of cases.

## 4. Discussion

A current recommendation in triaging EC is focused on testing the TP53, POLE, and MSI mutational status. Our study points out that the FGFR-2 mutation has no less meaning than TP53.

To the present day, the risk stratification systems of EC use the conventional rules of TNM classification as the depth of myometrial invasion, lymph node metastases, histology, and tumor grade. Genomic factors are not yet in standard clinical use for the assessment of prognosis. The subtypes proposed by Bokhman in 1983 are still widely accepted and used. The Cancer Genome Atlas funded by the National Cancer Institute provided us with a modern insight into the EC driving pathway and tumor biology [19]. However, the molecular panel we used lacks the POLE gene; we noted similar results. In our group, tumor grade correlated well with TP53 and FGFR2 mutation (*p* < 0.01). Interestingly, the FGFR2 mutation accelerated the stage progress. Moreover, in a separate analysis covering the number of a point mutation in a singular gene, we observed a marked impact of hypermutated TP53 (copy number high) on the clinical stage (*p* < 0.01).

The majority of EC can be classified into one of the four molecular subgroups. However, in a small subset (3–5%) of patients, the molecular analysis will show more than one classifying alteration (e.g., POLEmut-MMRd EC, POLEmut-p53abn EC, MMRd-p53abn EC, or POLEmut-MMRd-p53abn EC). As there are distinct prognostic differences between the four molecular subgroups, the question arises as to which biological behavior these multiple classifiers follow. This dilemma is most pronounced in the combination of POLEmut-p53abn EC. The tumor exhibits a favorable pathogenic mutation in the POLE exonuclease domain and the unfavorable aberrant p53 IHC expression [15,20,21]. It seems that the proposed four-tiered classification will evolve in the future. In our cohort, we noted 28% MSI frequency with a dominance of MLH-1 silencing. Microsatellite instability, in general, affected RFS (*p* < 0.01), but MLH-1 aberration to tumor stage and grade (*p* < 0.01). According to the mixed molecular subgroups, our study showed a coincidence of MSI with TP53 mutation in four cases, which means that about 20% of MSI cases were additionally TP53 mutated. Imboden et al. reported a large cohort analysis in aspect POLE mutated EC. They reported a significant POLE mutation heterogeneity with hotspots in c.857C > G and c.1231G > C and better overall survival POLE-mutated patients [22]. We mentioned ten EC cases with negative mutation results. These may be only POLE-mutated, and this needs further testing.

The most important clinical aspect is how to correctly segregate EC according to survival. The traditional three-tiered FIGO classification separates EC into three grades (G1, G2, and G3). The binary FIGO grading system was proposed close to 20 years ago, being in line with gastrointestinal pathology [23,24]. For repeatable classification, FIGO grade 3 endometrioid, serous, and clear cell carcinomas will be considered “high-grade endometrial carcinomas” (HGECs) and G1-G2 endometrioid carcinoma as low-grade endometrioid carcinomas (LGECs). The overwhelming number of LGECs belongs to the copy number low and MSI-H categories. Grade 3 endometrioid carcinomas are highly heterogenous, as they are found in every genomic category and are least represented in the copy number low group. Endometrioid carcinomas mostly have an endometrioid genomic profile, with or without TP53 mutation/high copy number alterations, whereas serous carcinomas have TP53 mutations/high copy number alterations without additional mutations characteristic of endometrioid carcinomas [25,26,27]. In our study, the TP53 mutation frequency was 20%, but some clinical significance we observed in the case of FGFR2 mutation (frequency 14%).

Going forward with the International Society of Gynecological Pathologists recommendation, we agree that TCGA classification should replace Bokhman’s. Unfortunately, there are unresolved financing issues be updated by FGFR-2, in our opinion with no less meaning. At this moment, the immunohistochemistry of p53 and MSI is recommended together with POLE-gene sequencing.

Assuming 6.3% frequency of the POLE hotspot mutation with an excellent prognosis the molecular testing is helpful in correct patients triage [22]. Moreover, the research based on the cell cultures confirms our results concerning the contribution of stromal fibroblasts in epithelial-mesenchymal transition and following poor outcomes [26]. Being aware of serious restrictions of immunohistochemistry, especially p53, broad testing is only one way to triage patients to different treatment algorithms [25,26,27]. We recommend a larger genes panel and NGS as a first choice method.

## 5. Conclusions

Our research confirmed a substantial value of EC molecular sub-grouping. The present study unveiled a number mutated genes contributing to cancer development and progress. It seems that the original TCGA rules should be replaced by new ones, in our opinion with no less meaning. FGFR-2 plays an important role in cancer progress even if via the epithelial-mesenchymal transition. Looking from a clinical point of view, to apply better chemotherapy regiment, and finally to modify the patient surveillance [18,19,28,29].

Future studies in larger populations with a vast panel of genes are necessary to develop evidence-based recommendations. Although there are limited evidences of FGFR-2 contribution in EC genesis, we recommend continuing along those lines.

This study was limited by the select population of one region in Poland, and the focused follow-up time varied amid the same participants. In future research we plan to add a control group. Moreover, our genes panel did not cover the POLE gene. The strengths of this work include the long-term observation and the modern methods of genetic testing. Our results unveiled other driving mutations to compare with the recommended panel.

## Figures and Tables

**Figure 1 diagnostics-12-01737-f001:**
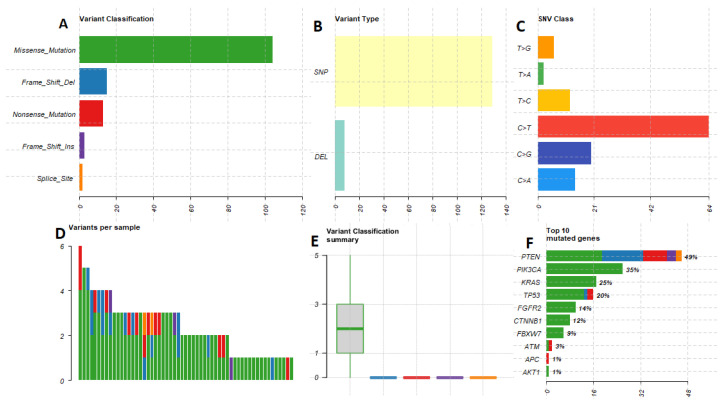
A graphical presentation of molecular findings. (**A**) comparison the mutation types with an evident dominance of missense mutation. (**B**) We noted the single nucleotide polymorphism as a main molecular aberration. (**C**) The cytosine-thymidine shift was observed to be the most frequent. (**D**) The graph presents several mutations in studied cases with emphasis on mutation variants. (**E**) Number of discovered mutations per case with a key role of missense type. (**F**) The variants according to gene. The PTEN gene presents a wide mutation landscape.

**Figure 2 diagnostics-12-01737-f002:**
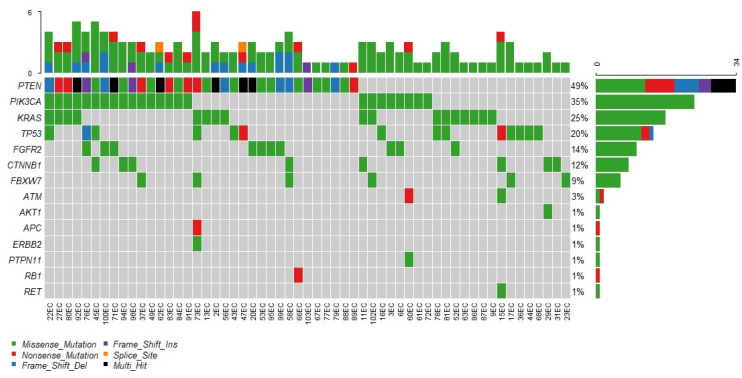
The occurrence and overlapping of additional molecular events. A picture illustrates the main pathways involved in EC genesis. We observed the PIK3CA-PTEN pathway with a frequency close to 50%. TP53 silencing was observed in 20% and the Wnt-pathway in 12%.

**Figure 3 diagnostics-12-01737-f003:**
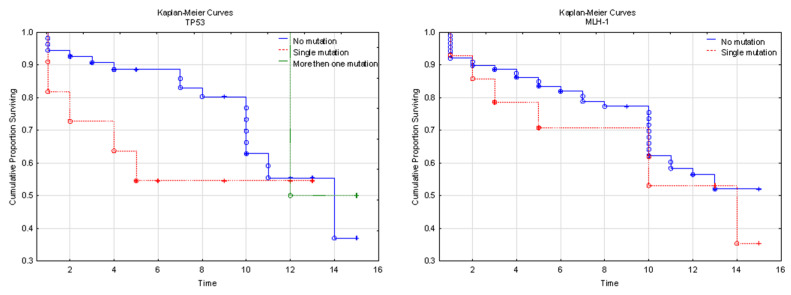
The Kaplan-Meier curves. We noticed only the strong impact of FGFR-2 mutation on OS (*p* < 0.01). A visualization of discovered mutations and their frequency according to the studied case. All cases of beta-catenin contribution overlaps with the AKT-mTOR Pathway.

**Figure 4 diagnostics-12-01737-f004:**
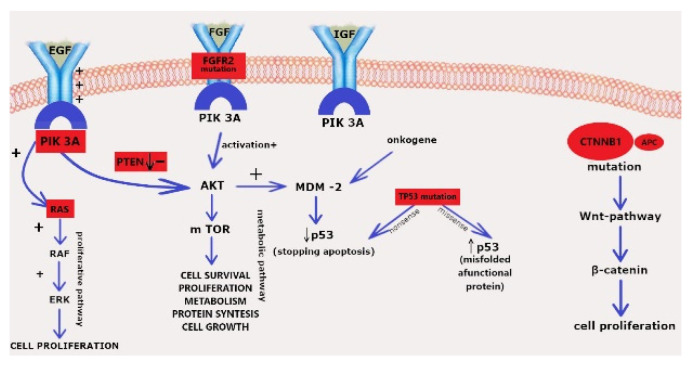
The main pathways involved in carcinogenesis in the studied group. The Kaplan-Meyer curves for TP53, MLH-1, MSI, and FGFR-2. TP53 χ^2^ = 3,90545 *p* = 0.14190, MLH-1 log-rank *p* = 0.53799, MSI log-rank *p* = 0.14569, and FGFR-2 log-rank *p* = 0.07095.

**Table 1 diagnostics-12-01737-t001:** General characteristics of the studied cohort.

	N	Average	Median	SD	Minimum	Maximum
Age (years)	103	71.08	70	10.98	48.00	99.00
<50	2	49	49	1.41	48	50
51–60	16	56.69	57	2.70	52	60
61–70	34	65.59	65.5	2.88	61	70
71–80	30	75.43	75	3.17	71	80
81–90	15	83.93	83	2.22	81	89
>90	6	94	93.5	2.68	91	99
BMI	103	34.07	34.20	2.44	23.00	38.50
18.5–24.99	2	23.9	23.9	1.27	23	24.8
25–29.99	1	29.8	29.8	-	29.8	29.8
30–34.99	61	33.06	33	1.20	30	34.9
35–39.99	39	36.29	36.2	1.01	35	38.5
Follow up time (years)	103	9.97	11	4.48	1	16
Rtg-therapy	59	4938.98	4600.00	1132.59	4500.00	9200.00
OS (years)	102	8.23	9.00	4.42	1.00	15.00
RFS (years)	6	4.67	3.00	4.23	1.00	10.00

## Data Availability

Data will be available from the corresponding author upon reasonable request.

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
