# Peer review of "The Roles of TP53 and FGFR2 in Progress Made Treating Endometrial Cancer"

_diagnostics, 2022, doi:10.3390/diagnostics12071737_

Round 1

Reviewer 1 Report

First of all, congratulations on your work.

I would like to see a more elaborate and detailed conclusions chapter focusing on the clinical implications and how these findings could be beneficial in the treatment options for patients with EC.

Reviewer 2 Report

Dear Author’s 
I was pleased to review your paper. It is a very interesting topic but in my opinion a manor revision is mandatory in order to improve your manuscrit.

1. please add the section discussion - is missing.

2. It will be more accurate if you take into consideration a control group.

3. Please explaine the novelty of your study.

4. Minor punctuation edits.

Reviewer 3 Report

In this study Authors have analyzed a vast panel of genes potentially involved in the genesis of endometrial cancer (EC) in a polish population of 103 white female patients with confirmed EC.

They performed sequencing using the Hot Spot Illumina panel and microsatellite stability with immunohistochemistry, confirming a key role of the TP53 mutation and parallelly some role of FGFR2 mutation.

They conclude that the correlation of FGFR2 mutation and TP53 (high copy number) in high-grade ECs is comparable with other studies.

The extent of the study is of some importance, even if localized to a limited geographical area. The reported results seem to confirm the meaning of FGFR2 mutation.

However, since FGFR2 belongs to pathways common to TP53, were these mutations of FGFR2 almost predictable and unsurprising?

Some errors in Fig.4: oncoqene; proliferafire, ..

In session 4. Discussion, a typo should have occurred.

I wonder if the bibliography is sufficiently reported, in particular regarding p53 and EC. For instance, Int J Gynecol Pathol. 2019 Jan; 38(Iss 1 Suppl 1): S123–S131;

Int J Mol Sci. 2019 Nov; 20(21): 5482;

Yano, M., et al. Mod Pathol 32, 1023–1031 (2019); etc.
